

# Criterion-related validity and reliability of the Urdu version of the patient health questionnaire in a sample of community-based pregnant women in Pakistan

John A. Gallis[1,2], Joanna Maselko[3], Karen O'Donnell[2,4], Ke Song[1,2], Kiran Saqib[5], Elizabeth L. Turner[1,2] and Siham Sikander[5,6]

[1] Department of Biostatistics and Bioinformatics, Duke University, Durham, NC, United States of America
[2] Duke Global Health Institute, Duke University, Durham, NC, United States of America
[3] Department of Epidemiology, Gillings School of Public Health, University of North Carolina at Chapel Hill, Chapel Hill, NC, United States of America
[4] Center for Child and Family Health, Durham, NC, United States of America
[5] Human Development Research Foundation, Islamabad, Pakistan
[6] Health Services Academy, Islamabad, Pakistan

Corresponding author
Siham Sikander,
siham.sikander@hsa.edu.pk

## ABSTRACT

**Background**. Depression is one of the most prevalent, yet unrecognized but treatable mental disorders in low and middle income countries (LMICs). In such locations, screening tools that are easy-to-administer, valid, and reliable are needed to assist in detecting symptoms of depression. The Patient Health Questionnaire (PHQ-9) is one of the most widely used depression screeners. However, its applicability to community-based settings of Pakistan is limited by the lack of studies examining its validity and reliability in such settings. The current study aimed to demonstrate the criterion-related validity and internal reliability of the Urdu version of the PHQ-9 in a sample of community-based pregnant women in Pakistan compared to a diagnostic clinical interview, the Structured Clinical Interview for DSM disorders (SCID), using data from a depression treatment cluster randomized trial in rural Pakistan.

**Methods**. Pregnant women in a rural, low income sub-district in Pakistan were approached between October 2014 and February 2016 and, after providing informed consent, screened for depression using the Urdu version of the PHQ-9, with a cutoff of ≥10 used to indicate significant depressive symptoms. Following the PHQ-9, the diagnostic module for current major depressive episode of the SCID was administered. We examined the psychometric properties of PHQ-9 compared to SCID as a gold standard, using sensitivity, specificity, and negative and positive predictive value to measure the criterion-related validity of the PHQ-9 as an indicator of symptoms of depression. We computed area under the receiver operating characteristic curve to determine diagnostic accuracy, and used Cronbach's alpha to assess internal reliability.

**Results**. A total of 1,731 women in their third trimester of pregnancy were assessed for major depressive disorder. Of these women, 572 (33%) met the cutoff for significant depressive symptoms on PHQ-9, and 454 (26%) were assessed positive for depression using the SCID. The sensitivity and specificity of PHQ-9 at a cutoff of ≥10 was 94.7% and 88.9%, respectively. The positive and negative predictive values were 75.2% and
97.9%, respectively; and the area under the curve was 0.959. Internal reliability, as measured by Cronbach's alpha, was 0.844.

**Discussion**. Valid and reliable screening tools to assist in detecting symptoms of depressive disorder are needed in low income settings where depressive disorders are highly prevalent. The Urdu version of the PHQ-9 has not been previously validated against a well-known assessment of depression in a community setting among pregnant women in Pakistan. This study demonstrates that the Urdu version of the PHQ-9 has acceptable criterion-related validity and reliability for screening for depressive symptoms in Pakistan among community-based pregnant women; and when the recommended cut-off score of ≥10 is used it can also serve as an accurate screening tool for major depressive disorder.

# INTRODUCTION

## Scientific and clinical background

Depressive disorder is a public health priority: It was estimated to be the 15th leading cause of global disability adjusted life years (DALYs) as of 2015, an increase from the 19th leading cause in 1990 (*Kassebaum et al., 2016*). Depressive disorder was also the second leading cause of years lived with disability (YLDs) in 2010, accounting for an estimated 8.2% of all YLDs (*Ferrari et al., 2013*). Depression is one of the most prevalent, yet unrecognized but treatable mental disorders in low and middle income countries (LMICs); easy to administer, valid and reliable screening tools that can assist in detecting depression will be useful in both clinical care and research in geographic areas with limited access to psychiatric professionals.

One of the most widely used screening tools for depressive disorder is the Patient Health Questionnaire (PHQ-9) (*Manea, Gilbody & McMillan, 2015*). The PHQ-9 was originally developed by Spitzer and colleagues in the mid-1990s, as part of the Primary Care Evaluation of Mental Disorders (PRIME-MD), a diagnostic tool containing modules on 12 different mental health disorders. The PHQ-9 is a tool specific for screening for depressive symptoms, within the mood module of the original PRIME-MD (*Kroenke, Spitzer & Williams, 2001*; *Spitzer et al., 1999*; *Spitzer et al., 1994*).

The PHQ-9 is based on the Diagnostic and Statistical Manual of Mental Disorders (DSM-IV) criteria of depressive disorder and has nine items with a score ranging from 0 to 3 for each item (0 = not at all, 1 = several days, 2 = more than half days, 3 = almost every day of a two-week recall period). The total score ranges from 0 to 27. Using standard cutoffs, a score of 5–9 indicates mild severity of major depression, 10–14 indicates moderate severity, 15–19 indicates moderately severe depression, while 20 or higher indicates severe major depression (*Kroenke, Spitzer & Williams, 2001*). The 10 point cut-off is most commonly used to indicate symptoms having reached clinically significant levels

requiring full diagnostic assessment (*Manea, Gilbody & McMillan, 2012*). For this study, we generate a continuous total score as well as grade the depressive symptoms using a range of cutoffs.

Over the years, the PHQ-9 has been validated in many settings and has shown to have good sensitivity and specificity when compared to diagnostic interviews (*Manea, Gilbody & McMillan, 2015*; *Moriarty et al., 2015*). The PHQ-9 has also been translated into myriad languages, with many translations validated (*Adewuya, Ola & Afolabi, 2006*; *Bian et al., 2011*; *Hanwella, Ekanayake & Silva, 2014*; *Huang et al., 2006*; *Lotrakul, Sumrithe & Saipanish, 2008*; *Wulsin, Somoza & Heck, 2002*). A number of recent studies on its validity and internal reliability, for screening for depression, and in monitoring treatment response have been published (*Chen et al., 2010*; *Lamers et al., 2008*; *Monahan et al., 2008*; *Titov et al., 2011*; *Wang et al., 2014*). As noted above, a score of ≥10 on PHQ-9 has been recommended for use as a diagnostic cut-off for possible major depressive disorder (*Kroenke, Spitzer & Williams, 2001*), although more recently researchers have recommended less strict adherence to any single cut-off (*Manea, Gilbody & McMillan, 2012*). A recent meta-analysis of 36 studies conducted in a range of settings but mainly in high-income countries found that the PHQ-9 cut-off of 10 had 78% (95% confidence interval [CI]: 70%–84%) sensitivity and 87% (95% CI [84%–90%]) specificity for detecting major depressive disorder in primary care and hospital settings (*Moriarty et al., 2015*), when compared against a "gold standard" such as the structured clinical interview for DSM disorders (SCID).

In Pakistan, the Urdu version PHQ-9 has been used in hospital and community settings as a screening tool for depressive disorder (*Fraz, Khan & Sikander, 2013*; *Gholizadeh et al., 2017*). Although criterion-related validity of an Urdu version of the PHQ-9 has been tested against the validated Urdu version Self Reporting Questionnaire (SRQ) among patients with coronary artery disease in an urban setting in Pakistan (*Gholizadeh et al., 2017*), the PHQ-9 has not been tested for criterion-related validity in a rural setting nor among pregnant or perinatal (that is, including a number of weeks after birth) women in Pakistan. Pregnancy or the entire perinatal period is an especially vulnerable period for women to develop mood disorders, and Pakistan has a very high burden of perinatal depression (*Fisher et al., 2012*; *Rahman, Iqbal & Harrington, 2003*; *Saeed, Humayun & Raana, 2016*). Therefore, having a valid screening tool is useful not only to assist in case identification but also, if compared to a diagnostic criterion tool, for impact evaluation of psychosocial interventions in this population.

There is debate around whether and when to screen adults for depression in primary care settings (*Thombs et al., 2012*; *Thombs & Ziegelstein, 2014*). The United States Preventive Services task force recommends screening only when adequate resources exist to manage the condition, while the UK and Canadian task forces recommend against it (*Thombs & Ziegelstein, 2014*). There is even less information on the appropriateness of screening in perinatal populations (*Thombs et al., 2014*). However, much of this research has focused on high income countries. In rural areas of LMICs where mental health services are virtually non-existent (such as in rural Pakistan), no clear recommendations exist. Benefits and drawbacks of screening for depressive disorder must be weighed considering the health

resources that do exist and the negative consequences associated with leaving populations undiagnosed in which the prevalence of depression is estimated to be about 25% (*Rahman, Iqbal & Harrington, 2003*; *Saeed, Humayun & Raana, 2016*).

The current study aimed at demonstrating the criterion-related validity and internal reliability of the Urdu version of the PHQ-9 in a sample of community-based pregnant women in Pakistan compared to a diagnostic clinical interview, the Structured Clinical Interview for DSM disorders (SCID) (*Spitzer et al., 1992*). The major depressive episode section of the SCID has previously been translated into Urdu and culturally adapted (*Rahman et al., 2008*), and has been extensively used cross-culturally in studies of depression among pregnant women (*Gorman et al., 2004*; *Nast et al., 2013*). The primary goal of this study is to determine if the Urdu version of the PHQ-9 provides an adequate tool for testing for depressive symptoms among pregnant women in rural settings in Pakistan by examining its internal reliability and its criterion-related validity relative to the SCID. Ethical approval for this study was obtained for this study from both U.S.-based (Duke University) and Pakistan-based (Human Development Research Foundation) institutional review boards (IRBs). In reporting the results of this study, we followed the Standards for Reporting of Diagnostic Accuracy Studies (STARD) 2015 guidelines (*Cohen et al., 2016*).

## MATERIALS & METHODS

### Study design and setting

The current study is part of the Thinking Healthy Programme Peer-Delivered (THPP) and Thinking Healthy Programme Peer-Delivered Plus (THPP+) cluster randomized trials (Duke University IRB approval #Pro00047609; Human Development Research Foundation IRB approval #IRB/2014/002 and #IRB/003/2016) (*Sikander et al., 2015*; *Turner et al., 2016*). The trials were conducted in the sub-district Kallar Syedan, Rawalpindi District, Pakistan. The sub-district has a population of about 200,000, and is representative of a typical low-socioeconomic rural area of Potohar Plateau situated in the north of the Punjab Province of Pakistan. Most families depend on subsistence farming, supported by earnings of one or more adult male members serving in the armed forces, or working as government employees, semi-skilled, or unskilled laborers in the nearby cities. The spoken and written language in the study area is Urdu. Male and female literacy rates are approximately 64% and 40%, respectively, in rural Punjab province according to the report from the most recent Demographics and Health Survey (*National Institute of Population Studies—NIPS/Pakistan & ICF International, 2013*). Mental health services are virtually non-existent in the sub-district.

The aim of these trials is to test the effect on perinatal depression of an extended and peer-delivered version of a previously successful intervention (*Rahman et al., 2008*). All pregnant women, registered with community health workers (called Lady Health Workers) of the sub-district Kallar Syedan, Rawalpindi District, Pakistan, were approached by study personnel and, after providing written (or witnessed, if illiterate) informed consent, screened for depression using the Urdu version of the PHQ-9. Following the PHQ-9, the diagnostic module for current major depressive episode of the Structured Clinical

Interview for DSM disorders (SCID) was administered by the same assessor within the same interview (*Spitzer et al., 1992*).

The study was conducted from October 2014 to February 2016, coinciding with screening and baseline data collection for the THPP and THPP+ trials (*Sikander et al., 2015*; *Turner et al., 2016*). All interviews were administered by study personnel at the households of the participants or at the house of the Lady Health Worker, whichever was more convenient for the pregnant women. The details of the trials and the recruitment criteria of its community-based depressed and non-depressed pregnant women can be found in the respective protocol papers (*Sikander et al., 2015*; *Turner et al., 2016*). In brief, to be eligible for screening, women must be married, in their third trimester of pregnancy, 18 years of age or older, and intend to remain in the study area for at least 1 year. Women are ineligible if they do not speak one of the study languages (Urdu, Punjabi or Potohari) or if they require immediate inpatient care for any reason. Target sample size for the THPP+ trial was determined based on the estimated effect size of the primary outcome between intervention and control arms at 36 months (*Turner et al., 2016*).

In total, 1,910 pregnant women across the sub-district were approached for consent and screening for depressive symptoms; 25 (1.3%) women refused screening while 154 (8.1%) were ineligible for enrollment. Of those ineligible, 110 (71.4%) were women in their 1st or 2nd trimester of pregnancy, eight (5.2%) were younger than 18 years old and not able to provide consent, 11 (7.1%) did not intend to remain in the study area, 22 (14.3%) did not speak one of the study languages, and three (1.9%) required immediate inpatient care. Thus, 1,731 community-based pregnant women in their 3rd trimester of pregnancy were screened for current major depressive episode using the Urdu version PHQ-9, which was followed by the SCID diagnostic tool during the same interview. Of these, 1,154 were retained for the baseline sample of THPP+ because of the sampling design, for which the goal was to recruit into the study every woman who was considered depressed (PHQ-9 $\geq$ 10), and approximately one out of every three non-depressed (PHQ-9 < 10) women screened, based on the estimated prevalence of depression in the population (*Sikander et al., 2015*; *Turner et al., 2016*). Criterion-related validity and internal reliability of the PHQ-9 were estimated in the full screening sample ($n = 1,731$). All members of the research team conducting the interviews had a master's degree in either psychology or behavioral sciences, and all of the team members had extensive experience of administering diagnostic interviews for depression within the same study area (*Maselko et al., 2015*; *Turner et al., 2016*). They were also supervised weekly by a trained and experienced psychiatrist.

### Test methods

The translated Urdu version of the PHQ-9 was used as the index test (*Multicultural Mental Health Resource Centre, 2013*). Prior to its use, some difficult Urdu words were replaced by simpler and more frequently used words carrying the same meaning. This was done to help address any potential issue of comprehensibility when administered in a low literacy rural population. For example, a difficult and not commonly used Urdu word for "feeling down" was changed to an easier Urdu word with the same meaning. This was done by a panel of mental health experts with extensive clinical and field experience of working

in low literacy populations. Comprehensibility was examined by pilot testing the PHQ-9 among 250 pregnant women prior to the start to the study. These women were also asked if any of the items were not understood. None of these 250 women were part of the trial screening sample, and no adjustments were made to the index test as a result of the pilot testing.

The translated Urdu version of the major depressive episode section of the SCID was used as the reference standard. The SCID was chosen as the reference standard because the Urdu version of this tool was previously translated and culturally adapted for a prior study in this region using a rigorous procedure (*Rahman et al., 2008*; *Rahman et al., 2008*). The SCID is a semi-structured interview that generates case vs non-case diagnosis of current major depressive episode by inquiring about the individual symptoms of depression. To make a diagnosis of depression at least five symptoms are needed, including depressed mood or loss of interest (*Spitzer et al., 1992*). The Urdu and English version of both the index and reference test are included as a Supplementary File.

As previously mentioned, a cutoff of 10 or above on the PHQ-9 was used to identify women with sufficient depressive symptoms to indicate a high probability of a depressive disorder. This is a commonly used cutoff (*Manea, Gilbody & McMillan, 2015*; *Moriarty et al., 2015*), and was chosen for the THPP and THPP+ trials because of its high positive predictive value for the diagnosis of depressive disorder in the (non-Pakistan) literature (*Sikander et al., 2015*). Evaluating the validity of the cutoff of 10 or above on the PHQ-9 was prespecified based on the trial design. We also examined the sensitivity and specificity of cutoffs ranging from five to 20 as an exploratory analysis. Since the same interviewer administered both the PHQ-9 and SCID, the assessors were not blinded to the results of the PHQ-9 when administering the SCID.

## Statistical analysis

In order to test the criterion-related validity and reliability of scores from the Urdu version of PHQ-9, we examined the psychometric properties of PHQ-9 using SCID as a gold standard, using the THPP and THPP+ screening dataset of 1,731 women. There was no missing data, as all items from the PHQ-9 and a diagnosis from the SCID were available on all women screened, and no index test or reference standard results were indeterminate. To verify the association between PHQ-9 and SCID, we assessed the criterion-related validity of PHQ-9 in Urdu in reference to the SCID by calculating sensitivity, specificity, positive predictive value (PPV), and negative predictive value (NPV) using cutoff values ranging from five to 20. Sensitivity and specificity of cutoffs were also visualized using a receiver operating characteristic (ROC) curve, from which was computed area under the curve (AUC). The internal reliability of the Urdu version of the PHQ-9 was assessed based on the value of the overall Cronbach's alpha. We also computed the Cronbach's alpha with each item deleted to further examine the inter-item consistency. Data were analyzed using Stata 15 (StataCorp, College Station TX, USA).

**Table 1 Baseline participant demographic characteristics, by PHQ-9 and SCID, $n = 1,154$** Demographic characteristics are from a 1:1 depressed/non-depressed sample (as determined by PHQ-9 $\geq$ 10) of $n = 1,154$ women.

| | PHQ-9 cutoff at 10 | | SCID current depressive episode | | Total (N = 1,154) |
|---|---|---|---|---|---|
| | Not depressed (N = 584) | Depressed (N = 570) | Not depressed (N = 712) | Depressed (N = 442) | |
| **Age (in years)** | | | | | |
| Mean (SD) | 26.4 (4.3) | 27.0 (4.8) | 26.5 (4.3) | 27.0 (4.8) | 26.7 (4.5) |
| Median (Q1, Q3) | 26.0 (23.0, 29.0) | 27.0 (24.0, 30.0) | 26.0 (24.0, 30.0) | 27.0 (23.0, 30.0) | 26.0 (23.0, 30.0) |
| Range | (18.0–40.0) | (18.0–45.0) | (18.0–40.0) | (18.0–45.0) | (18.0–45.0) |
| **Total # of children in the household** | | | | | |
| Mean (SD) | 2.5 (2.6) | 3.0 (2.7) | 2.6 (2.6) | 3.1 (2.7) | 2.8 (2.7) |
| Median (Q1, Q3) | 2.0 (1.0, 4.0) | 2.0 (1.0, 4.0) | 2.0 (1.0, 4.0) | 2.0 (1.0, 4.0) | 2.0 (1.0, 4.0) |
| Range | (0.0–20.0) | (0.0–21.0) | (0.0–20.0) | (0.0–21.0) | (0.0–21.0) |
| **PHQ-9 total score** | | | | | |
| Mean (SD) | 2.8 (2.5) | 14.7 (3.7) | 4.7 (4.7) | 15.1 (4.0) | 8.7 (6.7) |
| Median (Q1, Q3) | 3.0 (0.0, 4.0) | 14.0 (12.0, 17.0) | 3.0 (1.0, 7.0) | 15.0 (12.0, 18.0) | 9.0 (3.0, 14.0) |
| Range | (0.0–9.0) | (10.0–27.0) | (0.0–25.0) | (2.0–27.0) | (0.0–27.0) |
| **Grades woman has passed** | | | | | |
| None (0) | 63 (10.8%) | 107 (18.8%) | 83 (11.7%) | 87 (19.7%) | 170 (14.7%) |
| Primary (1–5) | 87 (14.9%) | 139 (24.4%) | 124 (17.4%) | 102 (23.1%) | 226 (19.6%) |
| Middle (6–8) | 108 (18.5%) | 107 (18.8%) | 129 (18.1%) | 86 (19.5%) | 215 (18.6%) |
| Secondary (9–10) | 167 (28.6%) | 126 (22.1%) | 191 (26.8%) | 102 (23.1%) | 293 (25.4%) |
| Higher secondary (11–12) | 63 (10.8%) | 46 (8.1%) | 74 (10.4%) | 35 (7.9%) | 109 (9.4%) |
| Tertiary (>12) | 96 (16.4%) | 45 (7.9%) | 111 (15.6%) | 30 (6.8%) | 141 (12.2%) |
| **Grades husband has passed** | | | | | |
| None (0) | 33 (5.7%) | 55 (9.6%) | 42 (5.9%) | 46 (10.4%) | 88 (7.6%) |
| Primary (1–5) | 45 (7.7%) | 67 (11.8%) | 62 (8.7%) | 50 (11.3%) | 112 (9.7%) |
| Middle (6–8) | 105 (18.0%) | 137 (24.0%) | 132 (18.5%) | 110 (24.9%) | 242 (21.0%) |
| Secondary (9–10) | 286 (49.0%) | 243 (42.6%) | 347 (48.7%) | 182 (41.2%) | 529 (45.8%) |
| Higher secondary (11–12) | 65 (11.1%) | 47 (8.2%) | 73 (10.3%) | 39 (8.8%) | 112 (9.7%) |
| Tertiary (>12) | 50 (8.6%) | 21 (3.7%) | 56 (7.9%) | 15 (3.4%) | 71 (6.2%) |

**Notes.**
The baseline sample of $n = 1,154$ is a subset of the screening sample of $n = 1,731$.

## RESULTS

### Participants

A total of 1,731 women were screened for inclusion in the study. Overall, women had a mean (standard deviation [SD]) PHQ-9 score of 6.8 (6.3) [range: 0–27] in the screening sample. Table 1 displays demographic information that was additionally available on the 1,154 women who were invited to participate in the study based on the study design (*Turner et al., 2016*). These women had a mean (SD) PHQ-9 score of 8.7 (6.7) [range: 0–27] and mean (SD) age of 26.7 (4.5) years. To utilize the most data, we only report validity and reliability results for the full screened sample ($n = 1,731$). If we used 10 or above as our cutoff for PHQ-9, approximately one-third of the women screened were considered to

**Table 2   PHQ-9 Score cutoffs by SCID reference standard; screening sample, $n = 1,731$.**

| PHQ-9 severity of depression | SCID depression status | | Total ($N = 1,731$) |
|---|---|---|---|
| | Depressed ($N = 454$) | Non depressed ($N = 1,277$) | |
| Minimal, 0–4 | 5 (1.1%) | 850 (66.6%) | 855 (49.4%) |
| Mild, 5–9 | 19 (4.2%) | 285 (22.3%) | 304 (17.6%) |
| Moderate, 10–14 | 202 (44.5%) | 112 (8.8%) | 314 (18.1%) |
| Moderately severe, 15–19 | 160 (35.2%) | 27 (2.1%) | 187 (10.8%) |
| Severe, 20+ | 68 (15.0%) | 3 (0.2%) | 71 (4.1%) |

be at least moderately depressive. However, there were relatively fewer women (26%) who were diagnosed with depression based upon the results of SCID. The distribution of PHQ-9 scores and SCID results among all women screened are shown in Table 2. In comparison with SCID, the moderate severity group (scoring from 10 to 14) had the largest inconsistency, since 112 (8.8%) women who were not depressed according to SCID had moderate depression severity according to PHQ-9 results.

### Test results

Criterion-related validity test results are reported in Table 3. The sensitivity and specificity of PHQ-9 for a cutoff of 10 or more were 94.7% (95% CI [92.2%–96.6%]) and 88.9% (95% CI [87%–90.6%]), respectively. The PPV and NPV at this cutoff were 75.2% (95% CI [71.4%–78.7%]) and 97.9% (95% CI [96.9%–98.7%]). The tradeoff between sensitivity and specificity at each cutoff is also shown in Fig. 1. The area under the curve (AUC) was 0.959 (95% CI [0.950–0.968]). According to the results in Table 3 and Fig. 1, a cutoff of 10 or more seemed to be a good choice for using PHQ-9 as a depression screener, given high sensitivity and specificity. However, any cutoff between nine or more and 11 or more appeared to have good sensitivity and specificity, with the estimated sensitivity and specificity values above 85%. In their systematic review, *Manea, Gilbody & McMillan (2012)* also found cutoffs in this range similar to the cutoff of 10 or more based on psychometric properties.

We assessed internal reliability based on the overall standardized Cronbach's alpha for all items together and alpha when each item is deleted. The overall standardized Cronbach's alpha was 0.844, indicating a high internal consistency. The alpha if item deleted ranged from 0.809 to 0.842, which suggested that internal consistency remained unaffected even if one of the items was removed. Results are shown in Table 4.

### DISCUSSION

Depressive disorders are highly prevalent and issues around screening and accurate diagnosis are still being debated (*Manea, Gilbody & McMillan, 2012*), with some countries recommending screening only under certain circumstances, and other countries recommending against routine screening in primary care (*Thombs et al., 2012*; *Thombs & Ziegelstein, 2014*). In rural areas of LMICs where mental health services are limited, the benefits and drawbacks of screening for depressive disorder have to be weighed considering

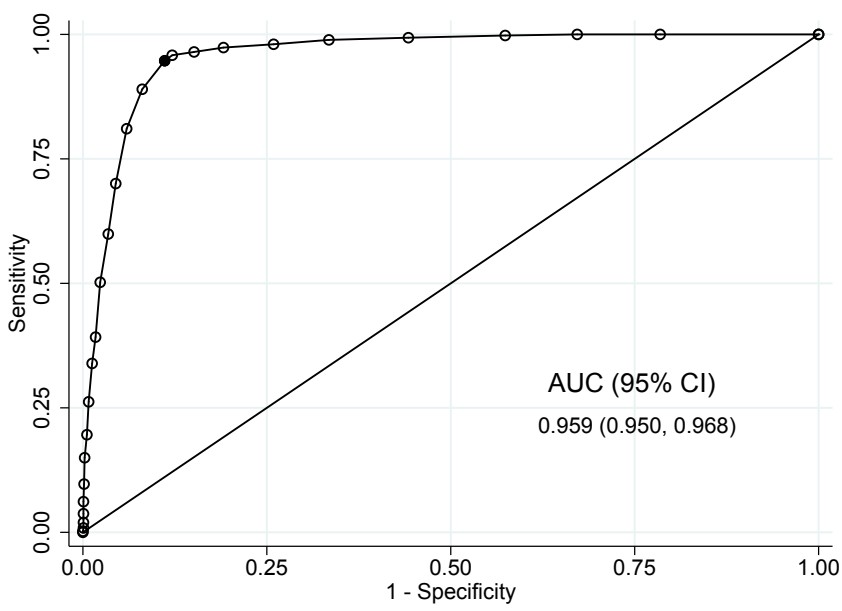

**Figure 1** **Receiver operating characteristic (ROC) curve comparing different cutoffs of PHQ-9 score to the SCID reference standard; screening sample, n = 1,731.** Abbreviation: AUC, area under the curve. Bolded point indicates the cutoff of ≥10.

**Table 3** **Psychometric properties of PHQ-9 at several cutoffs, with SCID as the reference standard; screening sample, n = 1,731.**

| PHQ-9 cutoffs | Sensitivity (95% CI) | Specificity (95% CI) | Positive predictive value (95% CI) | Negative predictive value (95% CI) |
|---|---|---|---|---|
| ≥5 | 98.9 (97.4–99.6) | 66.6 (63.9–69.1) | 51.3 (47.9–54.6) | 99.4 (98.6–99.8) |
| ≥6 | 98.0 (96.3–99.1) | 74.1 (71.6–76.5) | 57.3 (53.8–60.9) | 99.1 (98.2–99.6) |
| ≥7 | 97.4 (95.4–98.6) | 80.9 (78.6–83.0) | 64.4 (60.7–68.0) | 98.9 (98.0–99.4) |
| ≥8 | 96.5 (94.3–98.0) | 84.9 (82.8–86.8) | 69.4 (65.7–73.0) | 98.5 (97.6–99.2) |
| ≥9 | 95.8 (93.5–97.5) | 87.9 (85.9–89.6) | 73.7 (70.0–77.2) | 98.3 (97.4–99.0) |
| **≥10** | **94.7 (92.2–96.6)** | **88.9 (87.0–90.6)** | **75.2 (71.4–78.7)** | **97.9 (96.9–98.7)** |
| ≥11 | 89.0 (85.7–91.7) | 91.9 (90.3–93.4) | 79.7 (75.9–83.1) | 95.9 (94.6–97.0) |
| ≥12 | 81.1 (77.1–84.6) | 94.0 (92.6–95.3) | 82.9 (79.1–86.3) | 93.3 (91.8–94.6) |
| ≥13 | 70.0 (65.6–74.2) | 95.5 (94.3–96.6) | 84.8 (80.8–88.3) | 90.0 (88.2–91.5) |
| ≥14 | 59.9 (55.2–64.5) | 96.6 (95.4–97.5) | 86.1 (81.8–89.7) | 87.1 (85.3–88.8) |
| ≥15 | 50.2 (45.5–54.9) | 97.7 (96.7–98.4) | 88.4 (83.8–92.0) | 84.7 (82.7–86.5) |
| ≥16 | 39.2 (34.7–43.9) | 98.3 (97.4–98.9) | 89.0 (83.8–93.0) | 82.0 (80.0–83.9) |
| ≥17 | 33.9 (29.6–38.5) | 98.7 (98.0–99.3) | 90.6 (85.2–94.5) | 80.8 (78.7–82.7) |
| ≥18 | 26.2 (22.2–30.5) | 99.2 (98.6–99.6) | 92.2 (86.2–96.2) | 79.1 (77.0–81.1) |
| ≥19 | 19.6 (16.0–23.6) | 99.5 (98.9–99.8) | 92.7 (85.6–97.0) | 77.7 (75.6–79.7) |
| ≥20 | 15.0 (11.8–18.6) | 99.8 (99.3–100.0) | 95.8 (88.1–99.1) | 76.7 (74.6–78.8) |

**Notes.**

All numbers are percentages; using data from n = 1,731 women screened for depression.
Bolded text highlights the psychometric properties at a cutoff of ≥10.
**Table 4 Cronbach's alpha (internal reliability) results; screening sample, $n = 1,731$.**

| PHQ-9 Items[a] | Cronbach's alpha if item deleted[b] |
|---|---|
| 1. Feeling tired or having little energy | 0.828 |
| 2. Poor appetite or overeating | 0.842 |
| 3. Trouble falling or staying asleep, or sleeping too much | 0.836 |
| 4. Moving or speaking slowly | 0.820 |
| 5. Trouble concentrating | 0.836 |
| 6. Little interest or pleasure in doing things | 0.816 |
| 7. Feeling down, depressed, or hopeless | 0.809 |
| 8. Feeling bad about yourself | 0.824 |
| 9. Thoughts that you would better off dead, or of hurting yourself | 0.839 |
| **Overall Cronbach's alpha** | **0.844** |

**Notes.**

[a] Questions were asked in a different order than is standard.

[b] Items were standardized.

the health resources that do exist and the dangers associated with leaving populations undiagnosed. Valid and reliable screening tools to assist in detecting depressive disorder are much needed in low income settings where depressive disorders are more prevalent, such as in rural Pakistan, and where access to full diagnostic work-up is extremely limited. These valid and reliable screening tools are especially needed as more low-cost, effective community-based interventions are implemented and scaled up in these rural areas (*Rahman et al., 2008*). While Urdu versions of several instruments for detecting depressive disorders have been validated in Pakistani settings, the Urdu version of the PHQ-9 has not been tested for criterion-related validity in a community setting in Pakistan (*Ahmer, Faruqui & Aijaz, 2007*; *Gholizadeh et al., 2017*).

Through piloting the PHQ-9 in Urdu, we found that it was easily understood by community-based pregnant women, who answered both somatic and psychological items with equal ease. Using a diagnostic interview for comparison, data from the full screening sample of 1,731 pregnant women showed the Urdu version of the PHQ-9 had good criterion-related validity, similar to that reported in other studies (*Bian et al., 2011*; *Williams et al., 2005*). The Urdu version PHQ-9 also had a high internal reliability (Cronbach's alpha = 0.844) in this sample. The internal reliability in the current study is comparable to what other studies report within varied populations with alpha value of 0.85–0.90 (*Adewuya, Ola & Afolabi, 2006*; *Hanwella, Ekanayake & Silva, 2014*; *Martin et al., 2006*).

We note that for a cutoff of $\geq 10$ the specificity is right in line with the pooled (from 36 studies) specificity obtained in *Moriarty et al. (2015)*, however, the sensitivity is well above the upper limit of the pooled 95% confidence interval reported. This could be because of the interviewer administered both the SCID and PHQ-9 in the current study or because the studies summarized in *Moriarty et al. (2015)* do not include populations culturally similar to Pakistan. Also, the studies reported in *Moriarty et al. (2015)* included mainly hospital or clinic samples, rather than a community-based population. The high value of the AUC

(0.959) in this study demonstrates diagnostic accuracy which is in line with a number of studies (*Hanwella, Ekanayake & Silva, 2014*; *Kroenke, Spitzer & Williams, 2001*; *Martin et al., 2006*).

The most reliable estimates (using a clinical interview rather than a screener) among antenatal women in rural Pakistan show a depression prevalence of 23–26% (*Rahman et al., 2004*; *Rahman et al., 2008*), in line with the estimate from our study based on SCID (26%). Linking results from the current study to this estimated prevalence, a cutoff of ≥10 for PHQ-9 (with sensitivity and specificity of 94.7% and 88.9%, respectively) in a population where 26% of pregnant women are depressed corresponds to approximately three out of four (75.2%) new cases identified by the PHQ-9 being true cases.

One of the limitations of our study was that the same research team administered PHQ-9 and SCID in a single phase design and were not blind to PHQ-9 scores of the study participants when they administered the SCID. This is a potential source of interviewer bias, since knowing the results of one test may influence the interviewer's assessment of the woman on the other test. However, the interviewers were not aware that the data being collected would contribute to a validation study. In addition, each interviewer administered the SCID to a wide range of PHQ-9 high and low scoring participants, thus minimizing any potential interviewer bias of not being blind to PHQ-9 scores.

Another limitation of and potential source of bias in this study is that these women were not asked for previous medical records or diagnosis of depression they may have had. Including women with a prior diagnosis of depressive disorder in our study could lead to a sample with increased severity and prevalence of depression, which may inflate the reported sensitivity of PHQ-9 in relation to the SCID (*Thombs et al., 2011*). However, in this rural Pakistan setting medical histories and records are generally not kept or maintained and if available are not considered reliable. Additionally, given the lack of mental health services in this area, the probability of the women in this study area having had a previous diagnosis of depression is vanishingly small. Finally, the percentage of women (26%) in the screened sample who were diagnosed with depression based upon the results of the SCID is in line with the reported prevalence of depressive disorder during pregnancy of 23–26% from studies conducted in the same area of rural Pakistan (*Rahman et al., 2004*; *Rahman et al., 2008*).

## CONCLUSIONS

This study demonstrates that the Urdu version of the PHQ-9 has acceptable criterion-related validity and reliability for screening for depressive symptoms in Pakistan among community-based pregnant women; and when the recommended cut-off score of ≥10 is used it can also serve as an accurate screening tool for major depressive disorder.

## ACKNOWLEDGEMENTS

We would like to acknowledge the team members working on these projects and for their contributions, namely, Tayyiba Abbasi, Ikhlaq Ahmad, Qurat-ul-Ain, Najia Atif, Amina Bibi, Samina Bilal, Sonia Khan, Rakshanda Liaqat, Anum Nisar, Atif Rahman, Maria Sharif,

Shaffaq Zufiqar, Ahmed Zaidi. We would also like to thank the editor and two reviewers for providing helpful comments which improved the final version of this manuscript.

### Funding

The current paper is based on studies funded by the National Institute of Child Health & Human Development (NICHD), US, under award R01 HD075875, the National Institute of Mental Health, USA, under award number U19MH95687 and the Human Development Research Foundation, Pakistan. The funders had no role in study design, data collection and analysis, decision to publish, or preparation of the manuscript.

### Grant Disclosures

The following grant information was disclosed by the authors:
National Institute of Child Health & Human Development (NICHD), US: R01 HD075875.
National Institute of Mental Health, USA: U19MH95687.
Human Development Research Foundation, Pakistan.

### Competing Interests

The authors declare there are no competing interests.

### Author Contributions

- John A. Gallis and Ke Song analyzed the data, prepared figures and/or tables, authored or reviewed drafts of the paper, approved the final draft.
- Joanna Maselko and Karen O'Donnell conceived and designed the experiments, contributed reagents/materials/analysis tools, authored or reviewed drafts of the paper, approved the final draft.
- Kiran Saqib performed the experiments, prepared figures and/or tables, authored or reviewed drafts of the paper, approved the final draft.
- Elizabeth L. Turner conceived and designed the experiments, analyzed the data, authored or reviewed drafts of the paper, approved the final draft.
- Siham Sikander conceived and designed the experiments, performed the experiments, contributed reagents/materials/analysis tools, prepared figures and/or tables, authored or reviewed drafts of the paper, approved the final draft.

### Human Ethics

The following information was supplied relating to ethical approvals (i.e., approving body and any reference numbers):

The study has been granted ethical approvals from the Human Development Research Foundation (RB reference #: IRB/2014/002 and IRB/003/2016) and Duke University (USA) (IRB approval number: 47609) Institutional Review Boards.

## Data Availability

The anonymized data set contains the 1,731 women in the screening sample of our cluster randomized controlled trial, including their data from the PHQ-9 and SCID. The code allows the reproduction of numbers reported in the tables and in the article, with the exception of baseline demographic information on 1,154 of the women, since this data did not contribute to the analysis but is only provided as summary information.

The dataset and code are provided in the Supplemental Files.

## Supplemental Information

Supplemental information for this article can be found online at http://dx.doi.org/10.7717/peerj.5185#supplemental-information.

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
