# Peer review of "Criterion-related validity and reliability of the Urdu version of the patient health questionnaire in a sample of community-based pregnant women in Pakistan"

_PeerJ, doi:10.7717/peerj.5185_

## Round 0.1 · original submission · Major Revisions

Thank you for submitting your article to PeerJ. I have now received two reviews and I would like to thank both reviewers for their thoughtful assessments of the manuscript. The reviews are appended below and I won’t reiterate all the comments. However, both reviewers have highlighted important issues that need to be addressed in any revision.

Please ensure you address all reviewer comments; however, I believe that the following comments warrant particular attention when revising your manuscript:

1) Both reviewers commented on the fact that the Urdu version of the PHQ-9 has been used in previous research, raising questions about the rationale for the current study. Reviewer 2 suggests framing the study with regard to screening for perinatal depression, and reframing the manuscript to more accurately reflect the sample.

2) Both reviewers also commented on the fact that you evaluated sensitivity and specificity using a standard cut-off point. The rationale for this needs to be directly addressed in any revision of the manuscript.

3) Reviewer 1 recommends using the STARD guidelines to inform the reporting of the findings, and I believe that this would substantially strengthen the manuscript.

4) As noted by Reviewer 1, the benefits of screening are contested, particularly in the context of primary care. This debate needs to be incorporated into the background and addressed as part of the study rationale. Again, Reviewer 2’s suggestion to emphasise the nature of the sample may be helpful in thinking about this. Reviewer 1 has provided some references and I would encourage you to consider these, and I would also recommend considering the earlier Thombs et al (2012) paper.

Thank you again for submitting your article to PeerJ. I hope the reviewers’ comments are helpful in revising your manuscript, and that the points above are useful in focusing your response.

·

Basic reporting

This is a generally well-conducted study that usefully extends knowledge concerning the psychometric characteristics of an accessible and well-used measure. The findings are based on a large sample and will enable researchers to use the PHQ-9 with Urdu-speakers with greater confidence and clarity.

Some relatively minor corrections to the reporting would assist accuracy, clarity and readability; although most aspects of the design, conduct and analysis appear adequate, there are some key study limitations that need to be highlighted, and the overall reporting must better consider the context of depression screening/ case finding. The paper would benefit from the authors' checking the STARD guidelines (e.g. http://bmjopen.bmj.com/content/6/11/e012799) and considering relevant areas for further detail.

In the introduction section, the authors rightly quantify the impact of depressive disorder, making appropriate use of most recent Global Burden of Disease findings.
I find the initial sentences a little confusing and unnecessary- and would recommend omitting the first clauses (‘’It was listed as the fourth-leading cause of global disability adjusted life years (DALYs) in 1990, and was projected to become the second-leading 57 cause by 2020 (Murray et al. 1996). Reanalysis of the Global Burden of Disease Study using more and updated data reduced this number, but still..).
I think that simply reporting current DAY rank is more appropriate – and maybe simply noting that the age adjusted ranking by DALY in 1990 1005 and 2015 has increased (from 19 to 15) - I would omit the % global increase in DALY details that you have provided. (http://www.thelancet.com/pdfs/journals/lancet/PIIS0140-6736(16)31460-X.pdf - Figure 2)
Rather than the material relating to prior DALY summary ranking, it could be useful to note that depressive disorder accounts for substantial years of life lived with a disability (YLD) - 8.2% of global YLDs in 2010, making this condition the second leading cause of YLDs.(Ferrari et al, 2013 http://journals.plos.org/plosmedicine/article?id=10.1371/journal.pmed.1001547).

In some places the supposed benefits and value of ‘depression screening’ appears over-emphasised. There is overlap and some shared usage for the terms screening and case-finding - however, the available evidence does not seem to indicate clear benefits/ clinical usefulness of screening for depression in unselected groups (see for instance Thombs, 2014 http://www.bmj.com/content/348/bmj.g1253?variant=full-text&hwshib2=authn%3A1521546726%3A20180319%253A1ab0944b-ed16-45ab-b338-ee9cddfd397e%3A0%3A0%3A0%3A4CXS4yD783sDTM15gKwsHQ%3D%3D – and you will be aware that Manea et al summarise the uncertainties about this in the introduction to their review).
So, I would recommend you modify some instances of the use of this term: the sentence - ‘…Therefore, having a valid screening tool is imperative not only for detection but, if compared to a diagnostic criterion tool, for impact evaluation of psychosocial interventions in this population..’ should be altered – maybe to note that - having a valid screening or case-finding tool will be useful to assist case identification, particularly in at-risk groups or in those patients where depression is suspected..
And earlier, where you note – ‘…easy to administer, valid and reliable screening tools that can detect depression are needed for both clinical care and research in geographic areas with limited access to psychiatric professionals. it might be better to slightly modify this to ‘..easy to administer, valid and reliable screening tools that can assist in detecting depression may be useful in clinical care as well as in research ..

Clearly you have ethical approval for the parent studies, and you have noted this in parentheses at the start of the study design section; it could be useful to note here (explicitly) that ethical approval has been provided.

The results section would be improved by omitting comparison with other findings (Previous studies performed in the same area as that of the current study have reported prevalence of depressive disorder during pregnancy to be around 23-26% (Rahman et al. 2004; Rahman et al. 2008) – best only to compare findings in the Discussion section.

The initial sentence of the Discussion section should provide a little more brief accurate detail about depression screening – not just that ‘..issues around screening and accurate diagnosis are still being debated..’ this context is of key importance to the study/ topic and merits a little more detail.

In the Discussion or Conclusion section, it might be useful to consider providing a demonstration of what the psychometric values mean – i.e. a sentence or so indicating that at cut-off x and with sensitivity/ specificity values x, in a population with depression prevalence x%, then x% of newly identified cases are true cases (and - it would be useful to consider presenting this with prevalence rates determined by diagnostic interview rather than the usually higher values evident from symptom self-reports, and with prevalence of the level you report possibly together with pooled estimate rates [such as http://ww1.cpa-apc.org/Publications/Archives/CJP/2004/february/waraich.pdf]).

Experimental design

In the methods section detailing participants - it is noted that ‘...The details of the trials and the recruitment criterion..’ – I am sure that this should be criteria (not criterion) , and it would be useful if brief summary inclusion criteria were provided, with note that full details are available in the prior publications. Eligibility criteria are of particular importance in appraising the value of diagnostic accuracy studies (see STARD item 6).
Some of this (participants) material would fit better in a procedure or ?test methods section or with a ‘setting’ sub heading (but see journal format for guidance).

In the initial part of the ‘test methods’ section, you provide detail of translation of the PHQ-9 to Urdu, - but previously in the introduction section you note that the Urdu translation has been used in research in Pakistan and cite the or more publications. It would be useful to consider and clarify the rationale for, and the extent of, the further translation work that has been undertaken- the current text indicates a forward backward procedure with expert panel and relatively large test sample. I am uncertain as to why this was necessary (consider and clarify whether and what were the inadequacies in the published Urdu translated version).

Validity of the findings

It would be useful to specify that your analysis was particularly focused on evaluating PHQ-9 at standard cut-points – the STARD guidelines note the importance of - Definition of and rationale for test positivity cut-offs or result categories of the reference standard, distinguishing prespecified from exploratory
(See Cohen et al, 2016 http://bmjopen.bmj.com/content/6/11/e012799 item 12)

You note in the Discussion section (and elsewhere) that the assessment procedure did not utilise any blinding or masking – the same individuals administered the reference standard as the index test, all within the same interview. This is a potential source of bias in the study and you should use this term (potential bias) when identifying this limitation.

Studies that evaluate the screening tests / diagnostic accuracy should ensure removal any people/ patients who already have a diagnosis of, or are receiving treatment, for depression. Because screening is designed to identify those patients who may have a condition, but are neither seeking treatment nor have had the target condition otherwise recognised, then including any patients who are already identified as cases will increased prevalence and severity of depression in the sample, and may inflate the reported sensitivity of the index test in relation to the reference standard. (See - Thombs et al, 2011 - Risk of bias from inclusion of patients who already have diagnosis of or are undergoing treatment for depression in diagnostic accuracy studies of screening tools for depression: Systematic review. BMJ 343: d4825. doi:10.1136/bmj.d4825. PubMed: 21852353.).
Again – you need to note and discuss the sample and recruitment in relation to this potential source of bias.

Reviewer 2 ·

Basic reporting

no comment

Experimental design

no comment

Validity of the findings

no comment

Additional comments

In this study the authors aimed to validate the Urdu version of the PHQ for use in community settings. The measure is validated against the gold-standard SCID. The authors used data collected from pregnant women as part of a larger study.

The authors note that the Urdu version of the PHQ has been used in hospital and community settings, and has been validated in clinical settings. This raises the question of why the authors would expect different results in their sample. The requirement for a valid tool to assess perinatal depression makes sense. Given the sample, the authors might like to frame their introduction with this rationale in mind. In line with this – the sample is not simply a rural community-based sample, but a sample of pregnant women in their 3rd trimester. The entire manuscript could be re-framed to more accurately reflect the sample.

Were any adjustments made as a result of the pilot testing with 250 women? Were these same women included in the final sample?

Through the first half of the paper the authors claim they analyse both the screening and baseline samples separately. As they themselves note in the results section, this seems nonsensical given it is predominantly the same sample of women. I suggest they restrict their description to only the screening sample throughout the manuscript.

Did the authors have any a priori guidelines for what they would consider to be acceptable sensitivity and specificity? Without this, the decision to accept 10 as the cut-off seems to be either arbitrary, or simply a confirmation that the existing cut-off is acceptable (as opposed to optimal). If the study is better conceptualised as a confirmation of the existing cut-off the manuscript should be re-framed to better reflect this.

---

## Round 0.2 · Minor Revisions

Thank you for revising your manuscript. I have now received responses from both of the reviewers (see below). Both reviewers have requested minor revisions.

Reviewer 1 has recommended that the STARD guidelines be explicitly mentioned and cited in the manuscript, and I agree that this would be a useful addition.

Reviewer 2 has again highlighted the sample of pregnant women and the implications of this for the both the rationale of the study and the generalisability of the findings. This is an important consideration, and I strongly agree with the recommendations of Reviewer 2 to emphasis this focus in the title, abstract, and the paper, and to justify the focus on pregnant women more clearly in the introduction.

·

Basic reporting

I am pleased to note that this re-submitted paper provides a well-written report of a worthwhile study, which will be of interest and value to academic and clinical colleagues.

Experimental design

The amended report provides clear detail of the approach and methods, which appear appropriate and involve sufficient rigour.
The study deign and procedures appear to conform to STARD guidelines - my one additional suggestion is that you make explicit mention in the manuscript that you have followed STARD guidelines in the reporting of this study.

Validity of the findings

No further comments

Additional comments

Thanks for responding so clearly and comprehensively to the suggestions.

Reviewer 2 ·

Basic reporting

Generally well-written.

Experimental design

The design is sound, but as currently presented the sample used does not match the research question. See comments below, and on previous review about suggested re-framing.

Validity of the findings

The validity of the findings is fine, but implications can only be applied to samples of pregnant women, not rural areas of LMICs.

Additional comments

The authors have made a number of recommended changes that improve the manuscript. However, while they do acknowledge that they specifically sampled a group of pregnant women, the entire manuscript as still framed as assessing depression in a community-based sample. I still believe a stronger emphasis on the need to assess depression in pregnant women specifically would strengthen the rationale for the study, which would then match the sample. Likewise, the results will not simply be "especially applicable to pregnant women"; they will only be applicable to pregnant women. In my view this focus needs to come through in the manuscript title, the abstract and the manuscript itself. Otherwise readers are either being misled about the purpose, and significance, of the study or, (worse), the study design does not match the research questions being posed.

---

## Round 0.3 · accepted · Accept

Thank you for your revised submission and your responses to the reviewers' comments. I am pleased to accept your manuscript for publication in PeerJ. I do have some very minor text edits that I would recommend you make for readability.

Throughout the document, whenever you refer to "the validity and the reliability of the PHQ in community-based pregnant women in Pakistan", I would recommend inserting "in a sample of" (for example in the title: Criterion-related validity and reliability of the Urdu version of the patient health questionnaire in a sample of community-based pregnant women in Pakistan)